# Subspace Chronicles: How Linguistic Information Emerges, Shifts and Interacts during Language Model Training

**Max Müller-Eberstein**⊘    **Rob van der Goot**⊘    **Barbara Plank**⊘▲⊞    **Ivan Titov**♜⊷

⊘ Department of Computer Science, IT University of Copenhagen, Denmark
▲ Center for Information and Language Processing (CIS), LMU Munich, Germany
⊞ Munich Center for Machine Learning (MCML), Munich, Germany
♜ ILCC, University of Edinburgh, United Kingdom
⊷ ILLC, University of Amsterdam, The Netherlands
`mamy@itu.dk, robv@itu.dk, b.plank@lmu.de, ititov@inf.ed.ac.uk`

## Abstract

Representational spaces learned via language modeling are fundamental to Natural Language Processing (NLP), however there has been limited understanding regarding *how* and *when* during training various types of linguistic information emerge and interact. Leveraging a novel information theoretic probing suite, which enables direct comparisons of not just task performance, but their representational subspaces, we analyze nine tasks covering syntax, semantics and reasoning, across 2M pre-training steps and five seeds. We identify critical learning phases across tasks *and* time, during which subspaces emerge, share information, and later disentangle to specialize. Across these phases, syntactic knowledge is acquired rapidly after 0.5% of full training. Continued performance improvements primarily stem from the acquisition of open-domain knowledge, while semantics and reasoning tasks benefit from later boosts to long-range contextualization and higher specialization. Measuring cross-task similarity further reveals that linguistically related tasks share information throughout training, and do so more during the critical phase of learning than before or after. Our findings have implications for model interpretability, multi-task learning, and learning from limited data.

## 1 Introduction

Contemporary advances in NLP are built on the representational power of latent embedding spaces learned by self-supervised language models (LMs). At their core, these approaches are built on the distributional hypothesis (Harris, 1954; Firth, 1957), for which the effects of scale have been implicitly and explicitly studied via the community's use of increasingly large models and datasets (Teehan et al., 2022; Wei et al., 2022). The learning dynamics by which these capabilities emerge during LM pre-training have, however, remained largely understudied. Understanding *how* and *when* the

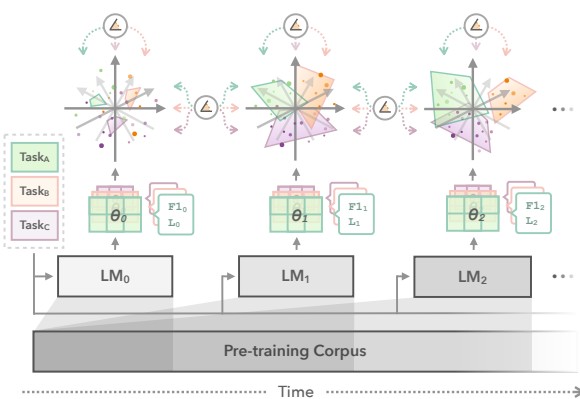

Figure 1: **Subspace Chronicles** via probes $\theta$ across LM training time, as measured by F1, codelength $L$, and subspace angles across tasks and time.

LM training objective begins to encode information that is relevant to downstream tasks is crucial, as this informs the limits of what can be learned using current objectives.

For identifying task-relevant information in LMs, probing has become an important tool (Adi et al., 2017; Conneau et al., 2018; Giulianelli et al., 2018; Rogers et al., 2020), which has already shown promise for revealing LM learning dynamics (Saphra and Lopez, 2019; Chiang et al., 2020; Liu et al., 2021). The predominant approach of quantifying linguistic information via task performance, however, misses important interactions at the representational level, i.e., whether actual linguistic information is present at any given training timestep (Hewitt and Liang, 2019; Voita and Titov, 2020), and how these representational subspaces change independently of model performance. Furthermore, performance alone does not indicate how different tasks and their related linguistic information interact with each other during training, and how much information they actually share.

By leveraging information-theoretic probes as characterizations of task-specific subspaces within

an LM's overall embedding space (Figure 1), we aim to answer these questions, and contribute:

- An information theoretic probing suite for extracting not just performance, but entire task-specific representational subspaces within an LM's embedding space, allowing us to measure changes to linguistic information over time *and* across tasks (Section 3).[1]

- A study of task subspace emergence, shifts and interactions across nine diverse linguistic tasks, 2M pre-training steps and five random initializations (Section 5).

- An analysis of these learning dynamics, which focuses on practical implications beyond performance to identify what can be learned given limited data, and how to effectively leverage this knowledge (Section 6).

## 2  Related Work

Although prior work has not specifically focused on the emergence of representational spaces, but rather on task performance, there has been an increased interest in the emergent capabilities of LMs. The community-wide trend of increasing model and dataset size has shown that certain skills (e.g., arithmetic) are linked to scale (Wei et al., 2022; Schaeffer et al., 2023), however Teehan et al. (2022) simultaneously identify a lack of work investigating skill emergence across the training time dimension.

To promote research in this direction, some projects have released intermediate training checkpoints. This notably includes MultiBERTs (Sellam et al., 2022) which studies cross-initialization variation in BERT (Devlin et al., 2019), as well as Mistral (Mistral, 2022), Pythia (Biderman et al., 2023) and BLOOM (Scao et al., 2023) which release checkpoints across training steps, seeds and/or pre-processing procedures.

In seminal work investigating learning dynamics, Saphra and Lopez (2019) measure how strongly hidden representations of a BiLSTM LM correlate with parts-of-speech (POS), semantic tags and topic information. By probing these characteristics across LM training time, they identify that POS is acquired earlier than topics. For Transformer models, Chiang et al. (2020) and Liu et al. (2021) probe intermediate checkpoints of ALBERT (Lan et al.,

---

[1]Code at https://github.com/mainlp/subspace-chronicles.

2020) and RoBERTa (Zhuang et al., 2021) respectively. They similarly identify that performance on syntactic tasks increases faster than on world knowledge or reasoning tasks, and that this pattern holds across pre-training corpora from different domains. This points towards consistent LM learning dynamics, however using only performance or correlations, it is difficult to interpret how the representation of this knowledge changes over time as well as how it overlaps across tasks.

Similar issues have arisen in single-checkpoint probing where seemingly task-specific information is identified even in random models, requiring control tasks (Hewitt and Liang, 2019), and causal interventions, such as masking the information required to solve a task (Lasri et al., 2022; Hanna et al., 2023). By using the rate of a model's compression of linguistic information, Voita and Titov (2020) alternatively propose an information theoretic measure to quantify the consistency with which task-relevant knowledge is encoded.

Another limitation of prior learning dynamics work is the inability to compare similarities *across* tasks—i.e., whether information is being shared. Aligned datasets, where the same input is annotated with different labels, could be used to measure performance differences across tasks in a more controlled manner, however they are only available for a limited set of tasks, and do not provide a direct, quantitative measure of task similarity.

By combining advances in information-theoretic probing with measures for subspace geometry, we propose a probing suite aimed at quantifying task-specific linguistic information, which allows for subspace comparisons across time as well as across tasks, without the need for aligned datasets.

## 3  Emergent Subspaces

To compare linguistic subspaces over time, we require snapshots of an LM across its training process (Section 3.1), a well-grounded probing procedure (Section 3.2), and measures for comparing the resulting subspaces (Section 3.3).

### 3.1  Encoders Across Time

For intermediate LM checkpoints, we make use of the 145 models published by Sellam et al. (2022) as part of the MultiBERTs project. They not only cover 2M training steps—double that of the original BERT (Devlin et al., 2019)—but also cover five seeds, allowing us to verify findings across multiple

initializations. The earliest available trained checkpoint is at 20k steps. Since our initial experiments showed that performance on many tasks had already stabilized at this point, we additionally train and release our own, more granular checkpoints for the early training stages between step 0–20k, for which we ensure alignment with the later, official checkpoints (details in Section 4.1).

## 3.2 Probing for Subspaces

Shifting our perspective from using a probe to measure task performance to the probe itself representing a task-specific subspace is key to enabling cross-task comparisons. In essence, a probe characterizes a subspace within which task-relevant information is particularly salient. For extracting a $c$-dimensional subspace (with $c$ being the number of classes in a linguistic task) from the $d$-dimensional general LM embedding space, we propose a modified linear probe $\theta \in \mathbb{R}^{d \times c}$ which learns to interpolate representations across the $l$ LM layers using a layer weighting $\alpha \in \mathbb{R}^l$ (Tenney et al., 2019). By training the probe to classify the data $(x, y) \in \mathcal{D}$, the resulting $\theta$ thus corresponds to the linear subspace within the overall LM which maximizes task-relevant information.

To measure the presence of task-specific information encoded by an LM, it is common to use the accuracy of a probe predicting a related linguistic property. However, accuracy fails to capture the amount of effort required by the probe to map the original embeddings into the task's label space and achieve the relevant level of performance. Intuitively, representations containing consistent and salient task information are easier to group by their class, resulting in high performance while requiring low probe complexity (and/or less training data). Mapping random representations with no class-wise consistency to their labels on the other hand requires more effort, resulting in higher probe complexity and requiring more data.

Information-theoretic probing (Voita and Titov, 2020) quantifies this intuition by incorporating the notion of probe complexity into the measure of task-relevant information. This is achieved by recasting the problem of training a probing classifier as learning to transmit all $(x, y) \in \mathcal{D}$ in as few bits as possible. In other words, it replaces probe accuracy with codelength $L$, which is the combination of the probe's quality of fit to the data as measured by $p_\theta(y|x)$ over $\mathcal{D}$, and the cost of transmitting the probe $\theta$ itself. The variational formulation of information-theoretic probing, which we use in our work, measures codelength by the bits-back compression algorithm (Honkela and Valpola, 2004). It is given by the evidence lower-bound:

$$
\begin{aligned}
L = -\, \mathbb{E}_{\theta \sim \beta} \left[ \sum_{x,y \in \mathcal{D}} \log_2 p_\theta(y|x) \right] & \\
+ \, \mathrm{KL}(\beta || \gamma) \,, &
\end{aligned}
\tag{1}
$$

where the cost of transmitting the probe corresponds to the Kullback-Leibler divergence between the posterior distribution of the probe parameters $\theta \sim \beta$ and a prior $\gamma$. The KL divergence term quantifies both the complexity of the probe with respect to the prior, as well as regularizes the training process towards a probe which achieves high performance, while being as close to the prior as possible. Following Voita and Titov (2020), we use a sparsity-inducing prior for $\gamma$, such that the resulting $\theta$ is as small of a transformation as possible.

In contrast to measuring task information via performance alone, codelength $L$ allows us to detect how consistently linguistic information is encoded by the LM (i.e., the shorter the better). At the same time, the linear transformation $\theta$ becomes an efficient characterization of the task-specific representational subspace. To further ground the amount of learned information against random representations, we use the probes of the randomly initialized models at checkpoint 0 as control values.

## 3.3 Subspace Comparisons

As each probe $\theta$ characterizes a task-specific subspace within the overall embedding space, comparisons between subspaces correspond to measuring the amount of information relevant to both tasks. To ensure that the subspaces extracted by our probing procedure are fully geometrically comparable, they are deliberately linear. Nonetheless, multiple factors must be considered to ensure accurate comparisons: First, matrices of the same dimensionality may have different rank, should one subspace be easier to encode than another. Similarly, one matrix may simply be a scaled or rotated version of another. Correlating representations using, e.g., Singular Vector or Projection Weighted Canonical Correlation Analysis (Raghu et al., 2017; Morcos et al., 2018) further assumes the underlying inputs

to be the same such that only the effect of representational changes is measured. When comparing across datasets with different inputs $x$ and different labels $y$, this is no longer given.

To fulfill the above requirements, we make use of Principal Subspace Angles (SSAs; Knyazev and Argentati, 2002). The measure is closely related to Grassmann distance (Hamm and Lee, 2008) which has been used to, e.g., compare low-rank adaptation matrices (Hu et al., 2022). It allows us to compare task-specific subspaces $\theta$ in their entirety and independently of individual instances, removing the requirement of matching $x$ across tasks. The distance between two subspaces $\theta_A \in \mathbb{R}^{d \times p}$ and $\theta_B \in \mathbb{R}^{d \times q}$ intuitively corresponds to the amount of 'energy' required to map one to the other. Using the orthonormal bases $Q_A = \text{orth}(\theta_A)$ and $Q_B = \text{orth}(\theta_B)$ of each subspace to compute the transformation magnitudes $M = Q_A^T Q_B$ furthermore ensures linear invariance. The final distance is obtained by converting $M$'s singular values $U\Sigma V^T = \text{SVD}(M)$ into angles between $0°$ and $90°$ (i.e., similar/dissimilar):

$$\text{SSA}(A, B) = \arccos(\text{diag}(\Sigma)) . \qquad (2)$$

We use SSAs to quantify the similarity between subspaces of the same task across time, as well as to compare subspaces of different tasks.

## 4 Experiment Setup

### 4.1 Early Pre-training

**Model** MultiBERTs (Sellam et al., 2022) cover 2M training steps, however our initial experiments showed that the earliest model at step 20k already contains close-to-final amounts of information for many tasks (see Section 5). This was even more pronounced for the early checkpoints of the larger LMs mentioned in Section 2. As such we train our own early checkpoints starting from the five MultiBERTs initializations at step 0. By saving 29 additional checkpoints up to step 20k, we aim to analyze when critical knowledge begins to be acquired during early training. To verify that trajectories match those of the official checkpoints, we train up to step 40k and compare results. Furthermore, we measure whether the subspace angles between the original and additional models are within the bounds expected for models sharing the same random initialization.

**Data** BERT (Devlin et al., 2019) is reportedly trained on 2.5B tokens from English Wikipedia (Wikimedia, 2022) and 800M tokens from the BookCorpus (Zhu et al., 2015). As the exact data are unavailable, Sellam et al. (2022) use an alternative corpus by Turc et al. (2019), which aims to reproduce the original pre-training data. Unfortunately, the latter is also not publicly available. Therefore, we gather a similar corpus using fully public versions of both corpora from the Hugging-Face Dataset Hub (Lhoest et al., 2021). We further provide scripts to re-create the exact data ordering, sentence pairing and subword masking to ensure that both our LMs and future work can use the same data instances in exactly the same order. Note however that our new checkpoints will have observed similar, but slightly different data (Appendix A.1).

**Training** Given the generated data order, the new LMs are trained according to Sellam et al. (2022), using the masked language modeling (MLM) and next sentence prediction (NSP) objectives. One update step consists of a batch with 256 sentence pairs for NSP which have been shuffled to be 50% correct/incorrect, and within which 15% of subword tokens (but 80 at most) are masked or replaced. For the optimizer, learning rate schedule and dropout, we match the hyperparameters of the original work (details in Appendix B.1). Even across five initializations, the environmental impact of these LM training runs is low, as we re-use the majority of checkpoints from MultiBERTs beyond step 20k.

**Pre-training Results** For LM training, we observe a consistent decrease in MLM and NSP loss (see Appendix C.1). As our models and MultiBERTs are not trained on the exact same data, we find that SSAs between our models and the originals are around $60°$, which is within the bounds of the angles we observe within the same training run in Figure 4 of Section 5.2. This has no effect on our later analyses which rely on checkpoints from within a training cohort. Surprisingly, we further find that although these models start from the same initialization, but are trained on different data, they are actually more similar to each other than models trained on the same data, but with different seeds. SSAs for checkpoints from the same timestep and training run, but across different seeds consistently measure $>80°$ (Figure 13 in Appendix C.1), highlighting that initial conditions have a stronger effect on representation learning than minor differences in the training data. Finally, our experiments in Section 5 show that our reproduced checkpoints align remarkably well with the official checkpoints,

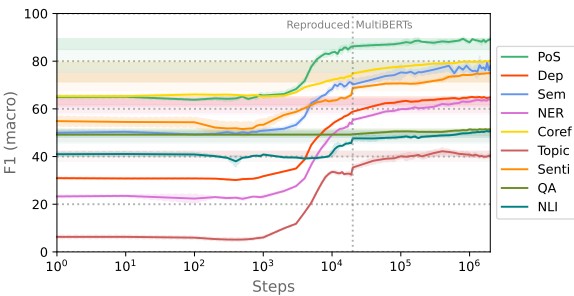

Figure 2: **F1 (macro) over LM Training Time** on each task's dev split (standard deviation across seeds). Dark/light shaded areas indicate 95%/90% of maximum performance. Reproduced checkpoints until 19k, Multi-BERTs from 20k to 2M steps.

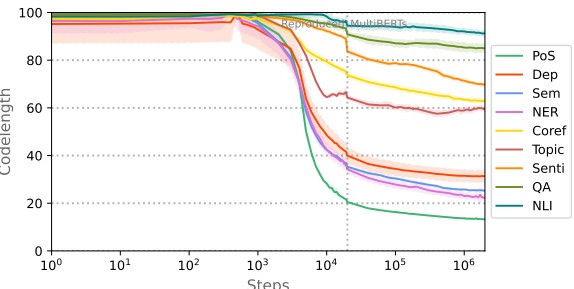

Figure 3: **Codelength Ratio over LM Training Time** as percentage of bits required to encode model and data with respect to the random model (standard deviation across seeds). Lower codelength corresponds to higher compression and more task-relevant information.

and we observe a continuation of this trajectory for the overlapping models between steps 20k and 40k.

## 4.2 Probing Suite

Given the 29 original MultiBERTs in addition to our own 29 early checkpoints, both covering five initializations, we analyze a total of 290 models using the methodology from Section 3. In order to cover a broad range of tasks across the linguistic hierarchy, we extract subspaces from nine datasets analyzing the following characteristics: parts-of-speech (PoS), named entities (NER) and coreference (COREF) from OntoNotes 5.0 (Pradhan et al., 2013), syntactic dependencies (DEP) from the English Web Treebank (Silveira et al., 2014), semantic tags (SEM) from the Parallel Meaning Bank (Abzianidze et al., 2017), TOPIC from the 20 Newsgroups corpus (Lang, 1995), sentiment (SENTI) from the binarized Stanford Sentiment Treebank (Socher et al., 2013), extractive question answering (QA) from the Stanford Question Answering Dataset (Rajpurkar et al., 2016), and natural language inference (NLI) from the Stanford Natural Language Inference Dataset (Bowman et al., 2015). Each task is probed and evaluated at the token-level (Saphra and Lopez, 2019) to measure the amount of task-relevant information within each contextualized embedding. In Appendix A.2 we further provide dataset and pre-processing details.

For each task, we train an information theoretic probe for a maximum of 30 epochs using the hyperparameters from Voita and Titov (2020). This results in 2,610 total probing runs for which we measure subword-level macro-F1, and that, more importantly, yield task subspaces for which we measure codelength, layer weights, and SSAs across tasks and time (setup details in Appendix B.2).

## 5 Results

Across the 2,610 probing runs, we analyze subspace emergence (Section 5.1), shifts (Section 5.2), and their interactions across tasks (Section 5.3). In the following figures, results are split between the official MultiBERTs and our own early pre-training checkpoints, which, as mentioned in Section 4.1, follow an overlapping and consistent trajectory, and have high representational similarity. Standard deviations are reported across random initializations, where we generally observe only minor differences in terms of performance.

## 5.1 Subspace Emergence

Starting from macro-F1 performance, Figure 2 shows clear learning phases with a steep increase between 1k and 10k update steps, followed by shallower growth until the end. Within subsets of tasks, we observe subtle differences: PoS and TOPIC appear to converge (i.e., >90% of final F1) within the 10k range. SEM follows a similar pattern, but has higher variance due to its many smaller classes. DEP and COREF, also gain most in the 1k–10k phase, but continue to slowly climb until converging later at around 100k steps. NER shares this slow incline and sees a continued increase even after 100k steps. SENTI and NLI also have early growth and see another small boost after 100k steps which the other tasks do not. Finally, QA also improves after 100k steps, however results are mostly around the 50% random baseline, as the linear probe is unable to solve this more complex task accurately. These results already suggest task groupings with different learning dynamics, however with F1 alone, it is difficult to understand interactions of the underlying linguistic knowledge.

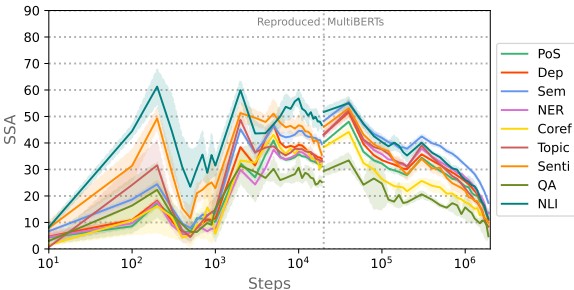

Figure 4: **Step-wise SSAs between Probes** indicating the degree of subspace change per update (standard deviation across seeds). Subspaces between original and reproduced checkpoints at step 20k are not comparable.

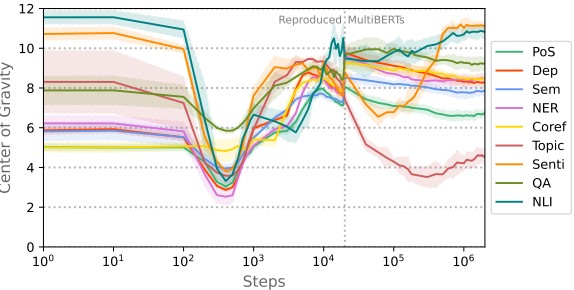

Figure 5: **Center of Gravity over Layers** measured following Tenney et al. (2019), across LM training time (standard deviation across seeds). For detailed weightings of all layers, refer to Appendix C.2.

Furthermore, even the random models at step 0 reach non-trivial scores (e.g., POS and COREF with >60% F1), as probes likely memorize random, but persistent, non-contextualized embeddings similarly to a majority baseline. This highlights the challenge of isolating task-specific knowledge using performance alone.

Turning to codelength in Figure 3, we measure the actual amount of learned information as grounded by the level of compression with respect to the random intialization. This measure confirms the dynamics from the performance graphs, however subspaces also continue changing to a larger degree than their performance counterparts suggest. Even after the 10k step critical learning phase POS information continues to be compressed more efficiently despite F1 convergence. In addition, codelength provides more nuance for COREF, QA and NLI, which see little performance gains, but for which subspaces are continuously changing and improving in terms of compression rate.

### 5.2 Representational Shifts

Both performance and codelength indicate distinct learning phases as well as continual changes to linguistic subspaces. Figure 4 plots these shifts by measuring subspace shifts as a function of SSAs from one timestep to the next. We observe particularly large updates after the first 100 steps, followed by smaller changes until the beginning of the steep learning phase at around 1k steps. During the large improvements between 1k and 10k steps, subspaces shift at a steady rate of around 45°, followed by a decrease to 5–20° at the end of training. Once again, subspaces are shifting across the entire training process despite F1 suggesting otherwise. Note that while learning rate scheduling can have effects on the degree of change, SSAs do not strictly ad-

here to it, exhibiting larger differences while the learning rate is low and vice versa. Figure 13 in Appendix C additionally shows how SSAs across checkpoints at the same timestep, but across different random seeds are consistently greater than 80°, indicating high dissimilarity. Compared to models starting from the same initialization, even if trained on slightly different data (i.e., reproduced and original), these angles indicate task subspaces' higher sensitivity to initial representational conditions than to training data differences.

Another important dimension for subspaces in Transformer-based LMs is layer depth. We plot layer weighting across time in Figure 5 using the center of gravity (COG) measure from Tenney et al. (2019), defined as $\sum_{i=0}^{l} \alpha_i i$. It summarizes the depth at which the probe finds the most salient amounts of task-relevant information. Appendix C.2 further shows the full layer-wise weights across time in greater detail. These weightings shed further light onto the underlying subspace shifts which surface in the other measures. Note that while SSAs across random initializations were large, variability of COG is low. This indicates that the layer depth at which task information is most salient can be consistent, while the way it is represented within the layers can vary substantially.

COG at step 0 essentially corresponds to the contextualization level of a task; memorizing non-contextualized embeddings at layer 0 or leveraging some random, but consistent mixing in the later layers. At the start of learning between 100 and 1k steps, all task subspaces dip into the lower half of the model. Together with the previously observed high initial subspace shift, this indicates that, beginning from the non-contextual embeddings in layer 0, contextualization becomes increasingly useful

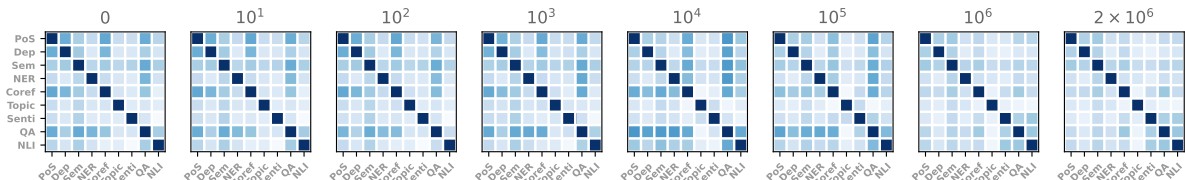

Figure 6: **Subspace Angles across Tasks** at start, end and at each order of magnitude of LM training time. Large angles (darker) and small angles (lighter) correspond to low and high similarity respectively.

from the bottom up. Paralleling the steep improvements of the other measures, CoG similarly climbs up throughout the model until the end of the critical learning phase in the 10k range. Until 500k steps, the layers for different tasks disentangle and stabilize. At this point syntactic and lower-level semantic tasks converge towards the middle layers. This specialization is especially prominent for TOPIC moving to the lower layers, while SENTI and NLI (and to a lesser degree QA), which require more complex intra-sentence interactions, move towards the higher layers. Recall that codelength and performance for SENTI, NLI and QA also improve around the same time. These dynamics show that the 'traditional NLP pipeline' in LMs (Tenney et al., 2019) actually emerges quite late, and that tasks appear to share layers for a large duration of LM training before specializing later on. Given that LMs are frequently undertrained (Hoffmann et al., 2022), these continual shifts to task-specific subspaces highlight that probing single checkpoints from specific timesteps may miss important dynamics regarding how tasks are represented with respect to each other.

### 5.3 Cross-task Interactions

The previous measures suggest that tasks could be grouped based on their shared learning dynamics. Our subspace-based approach for measuring cross-task SSAs (Figure 6) allows us to quantify this intuition. Comparing how much the subspaces of different tasks overlap provides a more holistic picture of task-relatedness than relying on specific data instances, and further shows how these similarities change over the duration of training.

Overall, angles between tasks follow linguistic intuitions: the syntax-related POS and DEP subspaces span highly similar regions in the general representational space. COREF, at least initially, also exhibits high similarity to the syntactic tasks, likely exploiting the same features to resolve, e.g., pronominal matches. The NER subspace overlaps

with POS and SEM, but less with DEP, potentially focusing on entity-level information, but less on the functional relationships between them. QA, NLI, and later SENTI, also share parts of their subspaces, while TOPIC is more distinct, having more overlap with the token-level NER and SEM tasks. While these patterns are already present at step 0, they become more pronounced during the early stages from 10k to 100k steps, and then become weaker towards the end as the subspaces disentangle and specialize within the model. Overall, subspaces appear to be related in a linguistically intuitive way, sharing more information during the critical learning phase, followed by later specialization.

These learning phases suggest that introducing linguistically motivated inductive biases into the model may be most beneficial during the early critical learning phase, rather than at the end, after which subspaces have specialized. This is an important consideration for multi-task learning, where the prevalent approach is to train on related tasks after the underlying LM has already converged. As it is also often unclear which tasks would be beneficial to jointly train on given a target task, our subspace-based approach to quantifying task similarity could provide a more empirically grounded relatedness measure than linguistic intuition.

## 6 Practical Implications

Based on these learning dynamics, we next analyze their impact on downstream applications. As the previous results suggest, around 10k–100k steps already suffice to achieve the highest information gains for most tasks and reach close to 90% of final codelength and probing performance. At the same time, performance and subspaces continue changing throughout training, even if to a lesser degree. In order to understand what is being learned in these later stages, we analyze finer-grained probe performance (Section 6.1), whether these dynamics are domain-specific (Section 6.2), and what effects they have on full fine-tuning (Section 6.3).

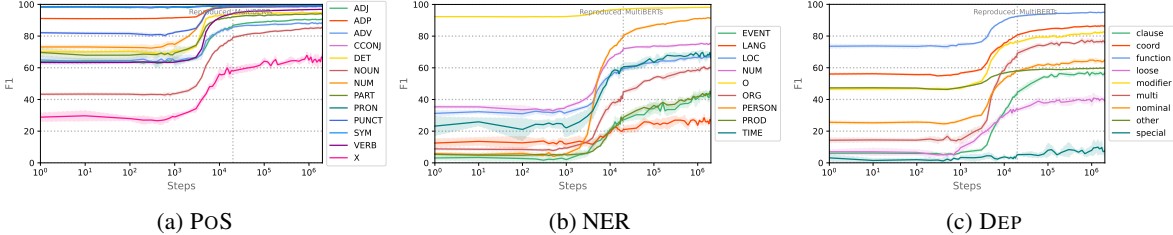

| (a) POS | (b) NER | (c) DEP |

Figure 7: **Class-wise F1 over LM Training Time** for POS, NER and DEP as measured on each task's dev split (standard deviation across seeds). For readability, classes are grouped according to Appendix A.2.

## 6.1 Class-wise Probing

First, we take a closer look at what is being learned during later LM training via class-wise F1. For POS (Figure 7a), most classes follow the general learning dynamics observed in Section 5, with no larger changes occurring beyond 10k steps. One outlier is the NOUN category, which continues to increase in performance while the other classes converge. Similarly for NER (Figure 7b), we observe that most classes stagnate beyond 10k steps, but that events (EVENT), products (PROD), and especially persons (PERSON) and organizations (ORG), see larger performance gains later on. What sets these classes apart from, e.g., determiners (DET) and pronouns (PRON), as well as times (TIME) and numbers (NUM), is that they represent open-domain knowledge. Performance on these classes likely improves as the LM observes more entities and learns to represent them better, while the closed classes are acquired and stabilize early on.

Turning to DEP, we observe similar continued improvements for, e.g., nominal relations, while functional has already converged. In addition, these class-wise dynamics further reveal that clausal relationships converge later than other relations, towards 100k steps. At the same time, modifier relations also see stronger gains starting at 100k steps. Both coincide with the performance and codelength improvements of QA, NLI and especially SENTI, as well as with the layer depth and subspace specializations. We hypothesize that this is another learning phase during which the existing lower-level syntactic knowledge is embedded in better long-range contextualization.

## 6.2 Domain Specificity

Next, we investigate whether the previous findings hold across domains, and whether LM training duration has an effect on cross-domain transferability by gathering out-of-domain (OOD) evaluation sets for five of the tasks. For POS, NER and COREF, we

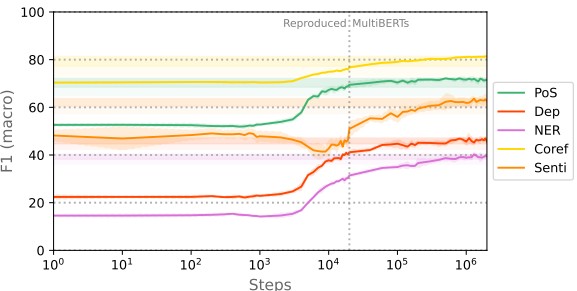

Figure 8: **OOD F1 (macro) over LM Training Time** with dark/light areas indicating 95%/90% of maximum performance (standard deviation across seeds).

split OntoNotes 5.0 (Pradhan et al., 2013) into documents from nw (newswire), bn (broadcast news), mg (magazines) for in-domain training, and bc (broadcast conversations), tc (telephone conversations), wd (web data) for OOD evaluation, based on the assumption that these sets are the most distinct from each other. In addition, we use English Tweebank v2 (Liu et al., 2018) for DEP, and TweetEval (Rosenthal et al., 2017) for SENTI. The learning dynamics in Figure 8 show that transfer scores are generally lower, but that previously observed trends such as the steep learning phase between 1k and 10k steps, and NER's steeper incline compared to POS and DEP continue to hold. SENTI sees an overall greater F1 increase given more training than in the in-domain case. From this view, there are however no obvious phases during which only OOD performance increases. Instead, the OOD tasks seem to benefit from the same types of information as the in-domain tasks.

## 6.3 Full Fine-tuning

Finally, in terms of downstream applicability, we evaluate the effect of pre-training duration on fully fine-tuned model performance. Due to the higher computational cost, we conduct a sweep over a subset of checkpoints for which the probing experiments indicated distinctive characteristics: Starting

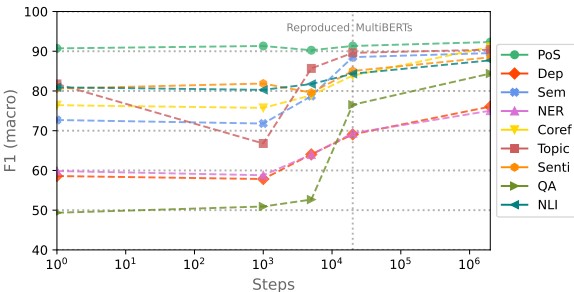

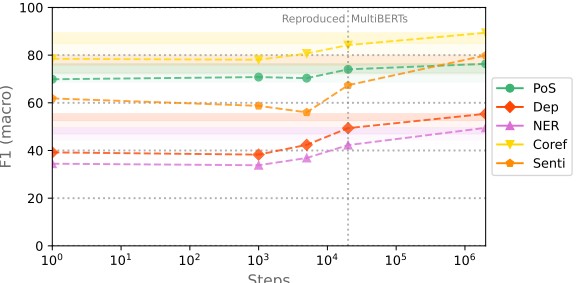

Figure 9: **F1 (macro) after Full Fine-tuning** of LMs initialized from checkpoints across LM pre-training time, as measured on each task's dev split (standard deviation across seeds too small to plot). Altered score range for readability.

Figure 10: **OOD F1 (macro) after Full Fine-tuning** of LMs initialized from checkpoints across LM pre-training time, as measured on each task's dev split (standard deviation across seeds too small to plot).

from scratch at 0 steps, at 1k steps before the critical learning phase begins, at 5k steps around the steepest incline, at 20k after most growth plateaus, and at the final 2M step checkpoint (training details in Appendix B.3). The results in Figure 9 show generally higher performance than for probing, as is to be expected. Interestingly, for the majority of tasks, full fine-tuning follows the same learning dynamics as probing suggests—exhibiting the greatest improvements in the 1k–20k range.

While performance for most tasks is within the 90–95% range at 20k steps, starting full fine-tuning from the point of steepest information gain at 5k steps does not seem to suffice to reach this target. Pre-training beyond 10k steps therefore seems crucial in order for the LM encoder to become beneficial. While it is possible to reach final performance on POS starting from a random encoder, even the similarly syntactic DEP sees substantial improvements up to the last pre-training step, likely connected to the improvement in longer-range contextualization observed in Section 6.1. Similar patterns can be observed for the other tasks, and especially for QA which only reaches usable performance beyond 20k steps. In terms of out-of-domain generalization for full fine-tuning, Figure 10 shows that continued pre-training does increase OOD performance, especially for more complex tasks. As such, LM pre-training, in combination with full fine-tuning, appears to be beneficial starting at around 10k steps for most traditional NLP tasks, followed by slower, but continued increases thereafter. As 10k updates constitute only 0.5% of full training, or 133M observed subword tokens, mid-resource languages may still benefit from language-specific, pre-trained LM encoders, even without access to English-level amounts of data.

For example, within the multilingual OSCAR corpus (Ortiz Suárez et al., 2020), at least 60 languages (∼40%) reach this threshold.

# 7 Conclusion

Our subspace-based approach to analyzing linguistic information across LM training has yielded deeper insights into their learning dynamics: In addition to the critical learning phase from 1k–10k steps, indicated by probing (Section 5.1) and fully fine-tuned performance (Section 6.3), our information theoretic approach identifies how linguistic subspaces continue changing, even if performance suggests otherwise (Section 5.2). For interpretability studies using single-checkpoint probing, this is crucial, as the information identified from a model may not be representative of final subspaces, especially if the model is undertrained (Hoffmann et al., 2022). Leveraging probes as characterizations of task-specific subspaces further allows us to quantify cross-task similarity, and surfaces how information is shared according to a linguistically intuitive hierarchy. This is particularly prominent during the critical learning phase, followed by later specialization, as more open-domain knowledge is acquired and the contextualization ability of the encoder improves. For multi-task learning, these dynamics imply that information sharing may be most effective early on, but more difficult after subspaces have specialized (Section 5.3). Finally, our analyses of OOD and full fine-tuning corroborate the previous learning dynamics (Section 6), showing that mid-resource languages could still benefit from pre-trained LM encoders at a fraction of the full fine-tuning costs, while simultaneously highlighting which information gains (i.e., open-domain, reasoning) require larger amounts of data.

## Limitations

Despite aiming for a high level of granularity, there are certain insights for which we lack compute and/or training statistics. Obtaining the highest resolution would require storing instance-level, second-order gradients during LM training, in order to identify how each data point influences the model (similarly to Achille et al., 2019). Neither our study, nor other checkpoint releases contain this information, however we release intermediate optimizer states from our own LM training to enable future studies at the gradient-level.

Another consideration is the complexity of our probes. To enable cross-task comparisons, we deliberately restricted our probes to linear models, as any non-linearities make it difficult to apply subspace comparison measures. As such they may not be able to capture more complex information, such as for QA and NLI. By using information theoretic probing, we are able to observe that task-relevant information is still being learned, albeit to a lower degree than for the other tasks. To nonetheless measure codelength for these more complex tasks, the same variational framework could be applied to non-linear models (Voita and Titov, 2020), however the resulting subspaces would also be non-linear, negatively impacting comparability.

As MultiBERTs forms our underlying LM architecture, more research is needed to verify whether the observed dynamics hold for larger, autoregressive LMs. While we believe that these larger models likely follow similar learning dynamics, extracting comparable subspaces will be even more difficult as scale increases. In initial experiments, we investigated checkpoints of the LLMs listed in Section 2, however similarly finding that performance had already converged at the earliest available checkpoint. Furthermore, checkpoints lacked information regarding how much data, i.e., subword tokens, had been observed at each step.

Finally, we would like to highlight that this study is correlatory, and that practitioners interested in a particular task should verify its specific learning dynamics by training LMs with targeted interventions (e.g., Lasri et al., 2022; Chen et al., 2023; Hanna et al., 2023). For this work, such interventions would have been out of scope, as we aim for wide task coverage to analyze interactions between them. Nonetheless, we hope that our findings on these learning dynamics can inform future task-specific studies of this type.

## Broader Impact

As this study examines the learning dynamics of LM representational spaces, its findings have wider downstream implications. Here, we specifically focus on three: First, for model interpretability, we identified that single-checkpoint probing does not provide the full picture with respect to how task-specific representations may change over time (e.g., layer depth in Section 5.2). This is critical when probing for sensitive information in LMs (e.g., bias), as representations may shift substantially, decreasing the effectiveness of interventions such as null-space projection (Ravfogel et al., 2020).

Second, we see a potential for multi-task learning to benefit from a better understanding of cross-task learning dynamics. Often it is unclear which task combinations boost/degrade each others' performance and what the underlying reasons are. Our findings suggest that similarities of task subspaces generally follow linguistic intuitions, but that there are distinct phases during which they share more or less information. Later specialization appears to be particularly important for more context-sensitive tasks, but may also make multi-task training more difficult, as tasks share less information at this stage. A question for future work may therefore be whether early-stage multi-task learning could lead to better downstream performance.

Third, for learning from limited data, this work identifies distinct phases during which different types of linguistic information are learned, as well as what their effects on fully fine-tuned and OOD performance are. We especially hope that the early acquisition of large amounts of information relevant for many traditional NLP tasks, encourages practitioners to revisit training encoders for under-resourced languages, despite the current trend towards larger models.

## Acknowledgements

Thanks to the ⊘ NLPnorth, ▲ MaiNLP and ♛ EdinburghNLP labs for their insightful feedback, particularly Joris Baan, Mike Zhang, Naomi Saphra, Verena Blaschke, Verna Dankers, Xinpeng Wang as well as to ITU's High-performance Computing Team. Additional thanks to the anonymous reviewers for their comments. This work is supported by the Independent Research Fund Denmark (DFF) Sapere Aude grant 9063-00077B and the ERC Consolidator Grant DIALECT 101043235.

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

# Appendix

# A Data Setup

## A.1 Language Modeling

**BookCorpus (Zhu et al., 2015)** was originally collected to train a sentence embedding model for aligning passages in books to scenes in their movie adaptations. It reportedly contains around 11k books in 16 genres, both stemming from the public domain as well as more contemporary works. In total, the corpus contains 74M pre-tokenized sentences. In our experiments, we use the public version from the HuggingFace Dataset Hub (Lhoest et al., 2021) which is available as bookcorpus.

**English Wikipedia (Wikimedia, 2022)** was used for training both original BERT (Devlin et al., 2019) as well as MultiBERTs (Sellam et al., 2022), however neither the version used, nor the pre-processing steps have been reported. In our experiments, we make use of the 20220301.en split of the wikipedia dataset from the HuggingFace Dataset Hub (Lhoest et al., 2021). It is available in a format where Wikipedia-specific markup has been removed, and each instance corresponds to the full text of an article. We further split these 6.5M articles into 144M sentences using spaCy v3.5.2 (Montani et al., 2023) and its en_core_web pre-processing pipeline.

**Pre-training Corpus** The final LM pre-training corpus consists of 109M sentence pairs from BookCorpus and Wikipedia, which are shuffled to be consecutive or randomly combined 50% of the time. They contain a total of 5.7B subword tokens of which 15% (and 80 at most) are replaced with a special [MASK] or another random token from the vocabulary. In practice, this results in 801M masked tokens, or 14.07% of the training data.

## A.2 Probing

**OntoNotes 5.0 (Pradhan et al., 2013)** contains documents from six domains, which we split into an in-domain and out-of-domain set for our analysis in Section 6.2. Each sentence is annotated with multiple layers of which we use: parts-of-speech (POS), named entities (NER), and coreference (COREF). It is split into 115,812 train, 15,680 dev, and 12,217 test instances.

POS follows the Penn Treebank schema (Marcus et al., 1993) with 51 classes. The class-wise analysis in Section 6.1 uses a mapping from this

labeling scheme to the Universal Part-of-Speech tagset (Petrov et al., 2012).

NER covers 18 entity types which are labeled using a `BIO` schema. For grouping these entities, we create the following custom mapping:

- **EVENT**: EVENT;

- **LANG**: LANGUAGE;

- **LOC**: GPE, LOC, NORP;

- **NUM**: CARDINAL, DATE, MONEY, ORDINAL, PERCENT, QUANTITY;

- **ORG**: ORG, FAC;

- **PERSON**: PERSON;

- **PROD**: PRODUCT; WORK_OF_ART, LAW;

- **TIME**: TIME.

COREF is built by extracting sentences with self-contained coreferences. The coreferring tokens are labeled as `I`, while all other tokens are labeled as `O`.

**English Web Treebank (Silveira et al., 2014)** contains syntactic dependencies (DEP) from 36 classes. To linearize the task, each token is labeled with the relation to its head word. For grouping the dependency relations, we use the nine categories from the official Universal Dependencies taxonomy (de Marneffe et al., 2014). The dataset consists of 12,543 train, 2,001 dev, and 2,077 test instances.

**English Tweebank v2 (Liu et al., 2018)** constitutes the OOD setup for DEP—specifically Twitter data. It adheres to the same Universal Dependencies relation label set as EWT, and consists of 1,639 (unused) train, 710 dev, and 1,201 test instances.

**Parallel Meaning Bank (Abzianidze et al., 2017)** contains three semantic annotation layers, of which we use the Universal Semantic Tags (SEM; Abzianidze and Bos, 2017). They denote 69 cross-lingual, lexical semantic categories at the token level. For grouping these tags, we combine the taxonomies of (Bjerva et al., 2016) and (Abzianidze et al., 2017) into 14 higher-level categories. The dataset consists of 7,745 train, 1,174 dev, and 1,053 test instances.

**20 Newsgroups (Lang, 1995)** contains email threads from 20 mailing lists, grouped by their TOPIC. Our experiments use the `bydate`-version which is sorted by date and removes duplicate entries and email headers containing the topic title. As the official data does not contain a dev split, we subdivide the training data, resulting in 9,051 train, 2,263 dev, and 7,532 test instance.

**Stanford Sentiment Treebank (Socher et al., 2013)** contains movie reviews along with their constituency parses and SENTI labels. We use the binarized SST-2 version with `positive/negative` labels. The dataset consists of 67,349 train, 872 dev, and 1,821 test instances.

**TweetEval (Rosenthal et al., 2017)** consitutes the OOD setup for SENTI. It contains seven annotation layers on Twitter data. We use the general sentiment labels and binarize them to match SST-2. The dataset consists of 45,615 (unused) train, 2,000 dev, and 12,284 test instances.

**Stanford Question Answering Dataset (Rajpurkar et al., 2016)** contains user-generated questions for which answer passages can be found in the corresponding Wikipedia articles, i.e., extractive question answering (QA). In our experiments, a question forms the first input sequence to the model, followed by a separator `[SEP]` token, and the relevant Wikipedia passage. Tokens within the answer passage are labeled as `I`, while everything else is `O`. The dataset consists of 87,599 train, and 10,570 dev instances.

**Stanford Natural Language Inference Dataset (Bowman et al., 2015)** contains premise-hypothesis pairs for natural language inference (NLI). Given a sentence pair, separated by `[SEP]`, the task is to predict whether the relation of the two inputs is an `entailment`, `contradiction`, or `neutral`. The dataset consists of 550,152 train, 10,000 dev, and 10,000 test instances.

## B  Experimental Setup

### B.1  Language Modeling

**Architecture**  The LM architecture used in our experiments is MultiBERTs (Sellam et al., 2022), which follows BERT$_{base}$ (Devlin et al., 2019), i.e., 12 layers with $d = 768$. We use the checkpoints published on HuggingFace Hub (Wolf et al., 2020) under `google/multiberts-seed_[0-4]-step_[0-2000k]`. For our own early checkpoints,

we start from `step_0` of each respective seed and train on the same data in the same order. From this training run, we store checkpoints and optimizer states at steps 10, 100–1,000 in increments of 100, 1,000–20,000 in increments of 1,000, and 40,000 for overlap comparisons, resulting in 29 additional models per initialization.

**Training** The LM training procedure for our own early checkpoints follows Sellam et al. (2022) as closely as possible. We use the AdamW optimizer (Loshchilov and Hutter, 2019) with a $10^{-4}$ learning rate, $\beta_1 = 0.9$, $\beta_2 = 0.999$ and a $10^{-2}$ weight decay. The learning rate is coupled to a polynomial schedule with a $10^4$ step warm-up and consequent decay until the end. Each batch contains 256 sentence pairs with a maximum length of 512 subword tokens. The model is trained to fill in masked tokens (MLM) using a language modeling head, as well as to predict whether one sentence follows another (NSP) based on the `[CLS]` token and a separate linear classification head.

## B.2 Probing

**Architecture** Each probe receives embeddings $\{h_0, \ldots, h_l\} \in \mathbb{R}^d$ from all $l$ layers (including the non-contextualized layer 0) as input, and summarizes them into a learned weighted average using $\alpha \in \mathbb{R}^l$ following (Tenney et al., 2019). This representation $h' = \sum_{i=0}^{l} \alpha_i h_i$ is then multiplied by a linear transformation $\theta \in \mathbb{R}^{d \times c}$ to produce logits for the $c$ output classes. Following the variational MDL formulation of Voita and Titov (2020), each parameter $w$ in $\theta$ is drawn from a normal distribution $w \sim \mathcal{N}(z\mu, z^2\sigma^2)$ with learned mean $\mu$ and variance $\sigma^2$, both scaled by $z$. There is one $z$ per input dimension $d$, which is is also drawn from a normal distribution $z \sim \mathcal{N}(\mu_z, \sigma_z^2)$ with its own learned mean $\mu_z$ and variance $\sigma_z^2$. During training this process is made differentiable using the reparametrization trick (Kingma et al., 2015). Each $w$ and $z$ pair is coupled to a joint normal-Jeffreys prior $\gamma(w, z) \propto \frac{1}{|z|}\mathcal{N}(w|0, z^2)$, according to Figueiredo (2001) and Louizos et al. (2017). It induces sparsity in $\theta$ by encouraging values of $w$ which are close to zero and have low variance. While the probe could theoretically be made more complex (i.e., non-linear), we specifically use a linear model in order to enable geometric comparisons between the resulting subspaces.

**Training** Both $\alpha$ and $\theta$ are jointly optimized by minimizing cross-entropy between predictions and gold labels. In addition, the KL divergence between $\theta$'s posterior $\beta$ and its sparsity inducing prior $\gamma$ are minimized according to Equation 1 to ensure maximum compression of both the data and the probe itself. Following (Voita and Titov, 2020), we use the Adam optimizer (Kingma and Ba, 2014) with a $10^{-3}$ learning rate, $\beta_1 = 0.9$, $\beta_2 = 0.999$ and 0 weight decay. Probes are trained with a batch size of 64 for a maximum of 30 epochs, and with early stopping on the development data if losses do not decrease.

**Evaluation** Following Saphra and Lopez (2019), we probe for task-specific information at the subword-level, meaning that for token-level tasks each token label is repeated across all of its constituent subwords, while for sequence-level tasks, the sequence label is repeated across all subwords. This corresponds to identifying task-specific information that is consistent across all contextualized embeddings within a sequence. To evaluate performance, we use macro-F1, as it is easier to interpret overall class-wise performance in-spite of class imbalances. E.g., NER and QA have a high number of 0 labels which are classified correctly with above 95% F1. With micro-F1, performance would appear unreasonably high, even if no named entities or answers would be identified.

## B.3 Full Fine-tuning

**Architecture** For the full fine-tuning experiments in Section 6.3, we train all parameters in the LM encoders from steps 0, 1k, 5k, 20k and 2M. For token-level tasks, we add a linear layer on top of the final contextualized embedding layer, while for sequence-level tasks, a linear layer is fed each input sequence's `[CLS]` token. These linear classification heads have the same dimensionality as the linear probes, but are not sampled variationally.

**Training** The fully fine-tuned models are trained using cross-entropy loss. Based on the recommendations in Devlin et al. (2019), we set the learning rate of the Adam optimizer (Kingma and Ba, 2014) to $3 \times 10^{-5}$, and retain the other hyperparameters. Models are trained for a maximum of 30 epochs, with early stopping on the development data when the loss stagnates.

## B.4 Implementation

Implementations use PyTorch v1.13 (Paszke et al., 2019) and NumPy v1.24 (Harris et al., 2020). Visualizations use matplotlib v3.6 (Hunter, 2007). Mod-

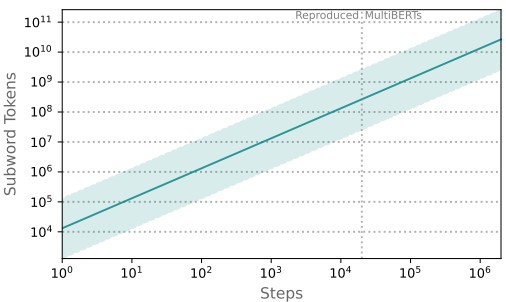

Figure 11: **Number of Subword Tokens Observed** during LM training, following the trajectory of our re-created dataset, plus the estimated upper/lower bounds for prior work.

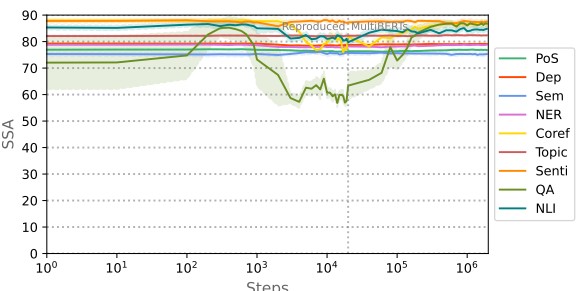

Figure 13: **SSAs across Random Initializations** for task-wise probes at each timestep (standard deviation across pairwise comparisons).

els were trained on an NVIDIA A100 GPU with 40GBs of VRAM and an AMD Epyc 7662 CPU. Probe training takes 10–60 minutes per checkpoint, dependent on the size of the dataset. Data pre-processing for language modeling (i.e., sentence splitting, tokenization, sampling and masking) takes 160 hours for the full dataset. The LM training process itself takes around 50 hours for 40k steps. The random seeds used in our experiments follow MultiBERTs (Sellam et al., 2022) and are: 0, 1, 2, 3, 4. Further, the code for reproducing our experiments is available at https://github.com/mainlp/subspace-chronicles.

## C  Additional Results

### C.1  Language Modeling

Figure 11 plots the number of subword tokens observed by the LM over the course of training. While this corresponds to the statistics of our re-created LM training corpus, the fact that the curve lies exactly between the feasible upper and lower-bounds given batch size and minimum/maximum subword tokens per sequence, we estimate that the original models also followed this trajectory.

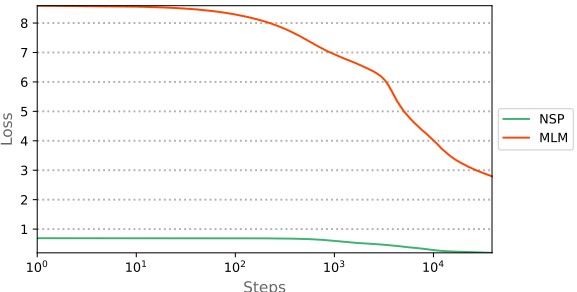

Figure 12: **MLM and NSP Losses** during the pre-training procedure described in Section 4.1, as measured for each batch from step 0 to 40k. Checkpoints, training statistics and optimizer states released on publication.

For our LM training runs from step 0 to 40k, Figure 12 shows how NSP and especially MLM losses start decreasing already after 100 pre-training steps. In general, losses on the actual LM pre-training tasks appears to decrease earlier by one order of magnitude, before probing performance and code-length improve.

Across LM training, task subspaces extracted at the same timestep, but across different seeds, are close to orthogonal to each other, as seen in Figure 13. This contrasts checkpoints starting from the same initialization (see Section 5.2), where the rate of change differs substantially across training, but generally remains under $60°$, even when trained on slightly different data (Section 4.1).

### C.2  Layer Weightings

In addition to the center of gravity (CoG) measure used in Section 5.2, the full layer weightings $\alpha$ for each task and timestep are shown in Figure 14. They provide a more detailed picture for cases in which multiple layers are weighted similarly, e.g., for the first checkpoints of DEP (Figure 14b, SEM (Figure 14c) and NER (Figure 14d). Both the earlier and later layers are weighted strongly, meaning that probes are making use of non-contextual plus mixed representations at these early stages. In contrast, SENTI and NLI almost exclusively rely on the last layer, while COREF and QA are initially spread out across all layer depths. Later on in training, these weightings also exhibit the specialization observed in Section 5.2, however weights are more spread out and do not collapse onto a single layer. This is most prominent with POS, but also DEP, SEM, NER and COREF, which all make use of information from across a wider range of depths.

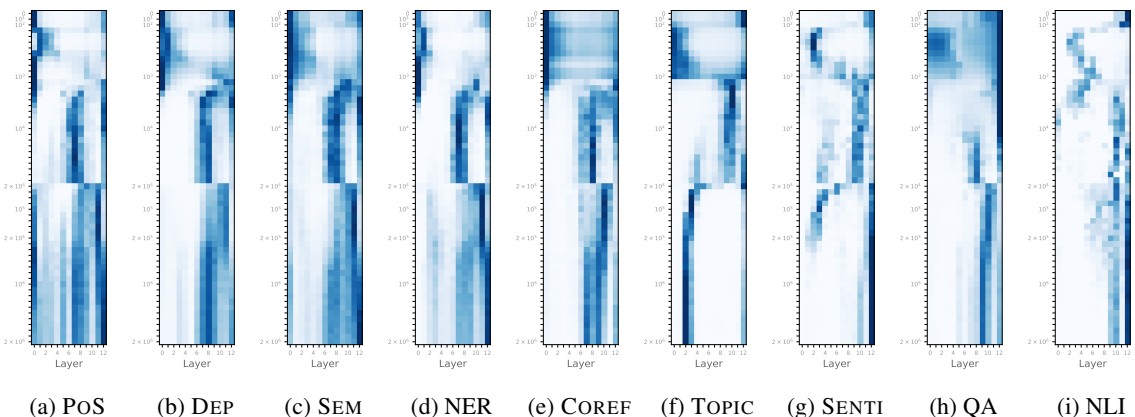

(a) POS  (b) DEP  (c) SEM  (d) NER  (e) COREF  (f) TOPIC  (g) SENTI  (h) QA  (i) NLI

Figure 14: **Layer Weightings $\alpha$ over LM Training Time** for all tasks, corresponding to the center of gravity measure in Section 5.2. Darker/lighter fields correspond to more/less weight respectively.