# OpenReview forum: "Subspace Chronicles: How Linguistic Information Emerges, Shifts and Interacts during Language Model Training"
_EMNLP/2023/Conference — EMNLP 2023 Findings_

### Official Review · Reviewer_29hu · 2023-07-31

**Soundness:** 4

**Excitement:**

3: Ambivalent: It has merits (e.g., it reports state-of-the-art results, the idea is nice), but there are key weaknesses (e.g., it describes incremental work), and it can significantly benefit from another round of revision. However, I won't object to accepting it if my co-reviewers champion it.

**Paper Topic And Main Contributions:**

This paper analyzes the representations of BERT-like models at the early stages of learning. The framing of the paper focuses on the subspace similarity method - comparing classifiers that decode information from the model's internal representations - but that analysis in fact ends up being a relatively minor part of the results.

There are some interesting observations about the timecourse of the emergence of linguistic information over the course of training, though nothing spectacular - for the most part, the models performance on the tasks increases most sharply between 1k and 10k steps, followed by a more gradual increase between 10k and 100k. This generally makes sense, because most linguistic tasks have an easy "head" and a hard "tail", so you can get pretty good accuracy with a not-very-strong model, but it's hard to get very good accuracy. This may also explain the (interesting) fact that there is more similarity across task subspaces in the earlier stages of training - there are coarse features that can get you pretty far on a lot of tasks.

The fact that the probe's compression rate improves at the later stages of training, even when F1 stays constant, is quite interesting (and seems vaguely related to the phenomenon of "grokking"). I thought better compression would also translate into better OOD performance but it doesn't seem to, at least not in the specific datasets used in this paper.

**Reasons To Accept:**

The experiments appear to be well done.

See previous box - there are some interesting findings.

**Reasons To Reject:**

I found the framing of the paper quite confusing - the authors don't make the novelty of the paper very clear. They describe earlier "probing" papers as identifying task-relevant information rather than "actual linguistic information", but in fact the papers that are cited as examples of the probing paradigm are ones where the goal is to decode linguistic categories such as the number of the subject (Giulianelli) or the depth of the three (Conneau), etc. This gets even more confusing when it turns out that many of the probes that the current papers use are "tasks" (such as NLI or QA) rather than "actual linguistic information" (such as POS tags).

I think the niche that this paper is filling isn't introducing tests for linguistic information - there are dozens of studies analyzing linguistic properties (as opposed to task performance) - but rather analyzing how linguistic abilities emerge over the course of training. But even this is something that has been done before, as the related work section lays out - what is the particular contribution of this paper? Doesn't the Saphra and Lopez study already study the emergence of linguistic properties such as POS tags over the course of training? Is this about LSTMs vs Transformers? If so, is this really the first paper to look at a transformer's learning dynamics? Or perhaps the novelty is in the fact that the authors looked at 5 seeds instead of just one?

The paper reports on a large number of relatively superficial analyses - it would have been nice to dig a little deeper into some of these analyses, and leave out the ones that have more obvious outcomes (e.g. the fact that performance for all tasks improves as you train on more data).

**Reproducibility:**

4: Could mostly reproduce the results, but there may be some variation because of sample variance or minor variations in their interpretation of the protocol or method.

**Reviewer Confidence:**

4: Quite sure. I tried to check the important points carefully. It's unlikely, though conceivable, that I missed something that should affect my ratings.

**Typos Grammar Style And Presentation Improvements:**

I couldn't follow the statement in the introduction that the distributional hypothesis has "scaled beyond semantics" - what does that mean?

---

> ### Author Rebuttal · Authors · 2023-08-28
>
> There are multiple contributions which underpin the findings of this paper, which we hope to clarify here: Saphra and Lopez (2019)’s work on LSTMs employs SVCCA, which requires representations to originate from the same input data. Cross-task analyses therefore require a single dataset to be annotated with multiple labels. This applies to all prior learning dynamics work, as the information learned for different individual properties are not directly comparable to each other.
>
> Our main contribution—and finding, as mentioned in your review—therefore lies in the ability to compare the linguistic information represented as subspaces of different tasks with different underlying datasets. This allows us to, for instance, identify the high amount of information sharing during the early stages of LM training (Section 5.3).
>
> With respect to your other questions, there are some prior works studying the learning dynamics of transformer models (Chiang et al., 2020; Liu et al., 2021), however they do not conduct cross-task analyses due to the aforementioned limitations, and do not cover multiple seeds of an otherwise identical LM architecture. The latter point, while not the main focus of our work, is nonetheless interesting to analyze, as we find that despite the different initializations' similar F1/compression trajectories, they are representationally quite dissimilar to each other (ll. 304–307).
>
> For OOD performance, we were also surprised to see that generalization ability does not necessarily increase during later stages. In separate experiments, we actually observed that this observation holds not just for our simple probing classifiers, but also for full fine-tuning.
>
> We hope that these points clarify our main contributions and highlight their novelty. Our findings focus less on the tests for linguistic information, but rather on its emergence over the course of training, as well as how it relates to each other across tasks.
>
> **Clarifications**
>
> > the distributional hypothesis has "scaled beyond semantics" - what does that mean?
>
> While language modeling is based on distributional semantics, it has far outgrown its only use as a method for learning dense representations for downstream tasks and is now used for all kinds of tasks in LLMs. We do agree that this phrasing is confusing, and would therefore opt to remove this from the paper’s final version.

---

### Official Review · Reviewer_USmm · 2023-08-03

**Soundness:** 3

**Excitement:**

3: Ambivalent: It has merits (e.g., it reports state-of-the-art results, the idea is nice), but there are key weaknesses (e.g., it describes incremental work), and it can significantly benefit from another round of revision. However, I won't object to accepting it if my co-reviewers champion it.

**Paper Topic And Main Contributions:**

The paper aims to explore how linguistic information in the representational space emerges, shifts and interacts during LM Training.
The main contribution is to propose a new probing suite to extract task-specific subspaces and compare them.

**Reasons To Accept:**

1. This work proposes a new probing suite to explore linguistic information during LM training and conducts some experiments to reveal some characteristics therein.

**Reasons To Reject:**

1. The experiments just conducts on a single model, without involving different model types and sizes, for which experimental results are much more needed.
2. The writing in this paper is not reader-friendly enough, e.g. section 3.2 and 3.3

**Reproducibility:**

4: Could mostly reproduce the results, but there may be some variation because of sample variance or minor variations in their interpretation of the protocol or method.

**Reviewer Confidence:**

4: Quite sure. I tried to check the important points carefully. It's unlikely, though conceivable, that I missed something that should affect my ratings.

---

> ### Author Rebuttal · Authors · 2023-08-28
>
> In addition to proposing "a new probing suite to extract task-specific subspaces" to reveal "some characteristics therein", we would like to highlight that this work sheds light on prior findings from learning dynamics at a finer-grained level of detail, using a stronger theoretical foundation (Reviewer n5w2). Our method of comparing representational subspaces across tasks allows us to identify critical phases of LM pre-training during which linguistic information is shared between tasks (Reviewer 29hu), leading to very different representations than prior work has assumed (e.g., layer depth in Tenney et al., 2019).
>
> > The experiments just conducts on a single model, without involving different model types and sizes, for which experimental results are much more needed.
>
> Regarding your primary feedback: While our experiments are based on a single but unique architecture, namely MultiBERTs, it is important to note that despite current trends, BERT-style embeddings are still widely used, and that our study is furthermore conducted on not just one model, but across five initializations and a total of 2.6k probing experiments. We do agree that similar analyses would be highly valuable for additional model types and sizes (see Limitations, ll. 666–679). For the scope of this paper, we however first decided to evaluate the feasibility of our new subspace-based probing methodology in higher detail and within our computational budget. Our initial experiments using larger architectures (mentioned in Sections 2, 3 and 4.1) showed that re-training all of these models would have been required in order to understand early-stage training dynamics. Even just for MultiBERTs, this would have required 9000 TPU hours (Sellam et al., 2022), so we hope that future work training LMs will release more of their earlier checkpoints.
>
> We would be glad to use an extra page to expand on the information theoretic methodology described in Sections 3.2 and 3.3. Since we do not wish to simply duplicate existing definitions from existing information theoretic probing work, any advice regarding how we may clarify specific parts would be greatly appreciated.

---

### Official Review · Reviewer_n5w2 · 2023-08-04

**Soundness:** 4

**Excitement:**

4: Strong: This paper deepens the understanding of some phenomenon or lowers the barriers to an existing research direction.

**Paper Topic And Main Contributions:**

This paper extends the previous work by using information-theoretic probes and examining the representation of transformers (across layer) during training. Building upon the learned probes, the author further measures the subspace expressed by the probes’ parameters.The main contribution of this work is that it gives a stronger theoretical support for previous conclusions. Encoding a linguistic property well means that it must be represented in a consistent and salient manner; a more organized subspace of learned representation is easier to group their classes, in contrast to a subspace of random representation.

**Questions For The Authors:**

Question A: is there any support for consistency and saliency in Line 165-167

Question B: when is the critical learning phase? Figure 2+3 have different critical learning phases than Figure 4+5.

Question C: could you clarify what do you mean by “a well-motivated, linguistically related set of tasks” (Line 484)? How do you expect researchers to make use of those tasks?

Question D: according to Fig 7c, the phrase seems to be around 10^4 (10k) instead of 100k?

Question E: Section 6.3 What is the Full Fine-tuning task?

**Reasons To Accept:**

This paper extends the previous work by using information-theoretic probes and examining the representation of transformers during training. This finer-grain and more theoretically supported (linear algebra analysis is theoretically more justified than accuracy) analysis provides stronger evidence even for a similar conclusion.

Despite gaps in Fig 4+5, it’s surprising to see that Figure 2+3 has a relatively smooth transition between reproduction and MultiBERT. This interesting evidence shows that the acquisition of linguistic property is similar at the same training stage.

**Reasons To Reject:**

The validity of putting BERT reproduction and MultiBERT in the same plot is questionable, as there are some gaps in Fig 4+5.

Section 6 seems to be a miscellaneous section that contains further analysis, and it seems hard to make connections to downstream and more practical tasks.

**Reproducibility:**

4: Could mostly reproduce the results, but there may be some variation because of sample variance or minor variations in their interpretation of the protocol or method.

**Reviewer Confidence:**

4: Quite sure. I tried to check the important points carefully. It's unlikely, though conceivable, that I missed something that should affect my ratings.

**Typos Grammar Style And Presentation Improvements:**

Figure 6 is confusing. There are too many small cells, so I can’t really see the dynamics of each cell. I wonder if it suffices to just have top-k most similar subspaces.

---

> ### Author Rebuttal · Authors · 2023-08-28
>
> To address your primary concern regarding the comparability of MultiBERTs and our own reproduced checkpoints: We would like to highlight that i) we carefully checked our early checkpoints to make sure they are comparable by looking at them in two ways (see next paragraphs) while avoiding costly full re-training, and ii) we leverage MultiBERTs as a unique resource.
>
> In more detail, we originally identified the need to retrain early checkpoints due to the earliest available checkpoints of MultiBERTs as well as multiple larger models listed in Section 2 (ll. 86–90) exhibiting close to the final levels of performance (see also Figures 2, 3, 7, 8 and 9). To do so within our available compute budget, we followed the original work as closely as possible in terms of data, architecture, hyperparameters (see Section 4), and by reaching out to original authors of MultiBERTs for advice.
>
> The resulting reproduced models were trained up to step 40k, beyond the intersection at 20k steps, in order to verify that performance trajectories overlap. This continued trajectory is visible in all F1 and code length related plots (Figures 2, 3, 7 8, 9). Continuing to train for another 10^6 steps with all five seeds would have required approximately 9000 TPU hours (Sellam et al., 2022), which we deemed unnecessarily wasteful. In this sense, MultiBERTs are a unique resource in that they provide many later checkpoints across multiple seeds, while being based on a base architecture for which we can reproduce earlier checkpoints.
>
> We additionally examined the representational similarity of the original and reproduced models (ll.289–311). This also relates to the gaps you mention for Figures 4 and 5. The subspace angles between the original and reproduced models are around 60 degrees (ll. 292–294), however this difference is comparable to angles observed within the same training run (see steps 10^2 and 10^3 in Figure 4). Furthermore, models from the same training run, but initialized using different seeds, have angles over 80 degrees (ll. 304–307), meaning that our reproductions are substantially closer to the original models than the same models are across seeds. Given additional space, and based on your suggestions, modifying the plots to show both the overlapping performance trajectories, as well as the subspace angles across different random initializations could be used to highlight the aforementioned observations.
>
> **Questions**
>
> > Question A: is there any support for consistency and saliency in Line 165-167
>
> Information which can be extracted easily (i.e., by a linear probe with high accuracy) can be seen as salient, and consistent representations for members of the same class lead to more efficiency in this process. Information theoretic probing and its measure of code length allow us to measure both properties in a more theoretically rigorous manner than using accuracy alone.
>
> > Question B: when is the critical learning phase? Figure 2+3 have different critical learning phases than Figure 4+5.
>
> We place the "critical learning phase" to be around 10^3–10^5 range, depending on the type of linguistic information. You are correct that the highest rate of change differs depending on the measure used. Sections 5.1, 5.2 and 5.3 therefore attempt to highlight different aspects of each.
>
> > Question C: could you clarify what do you mean by "a well-motivated, linguistically related set of tasks" (Line 484)? How do you expect researchers to make use of those tasks?
>
> In multi-task learning, it is frequently unclear which tasks boost or harm each other’s performance. Our experiments in Section 5.3 and Figure 6 show that task similarity in LM representations follow linguistic intuitions, and our additional findings regarding how this similarity changes over time (see also R3), can inform future work on training multi-task models from intermediate checkpoints or from scratch.
>
> > Question D: according to Fig 7c, the phrase seems to be around 10^4 (10k) instead of 100k?
>
> The largest rate of change is indeed at 10^4. We refer to the convergence of performance at around 10^5 (ll. 527–529).
>
> > Question E: Section 6.3 What is the Full Fine-tuning task?
>
> Since our probing experiments use simple classification models, which are regularized using the minimum description length objective (Section 3.2), our experiments in Section 6.3 use a standard full LM fine-tuning procedure to measure more realistic downstream performance.

---

### Meta-Review · Area_Chair_gcfh · 2023-09-19

**Recommendation:** 4

**Metareview:**

This paper presents a suite of probes for understanding when and where linguistic information (syntax, semantics, reasoning) is learnt in the embedded representations of MultiBERT models, including an information theoretic analysis supporting the specific formulation of the probes. The main concern shared by all reviewers is that the paper was often confusing and hard to follow. Some reviewers felt that treatment of highly related work was insufficient, some felt that the description of technical details was hard to follow, and some reviewers felt that the overall motivation/story of the paper was not clearly aligned with the provided results and analysis. If the paper is accepted, it should be substantially edited for clarity based on the feedback from the reviewers. After much clarification by the authors in the rebuttal, reviewers seemed to understand more, and found the results in the paper to be sound. While reviewers agreed that the topic was interesting and relevant, reviewers were not unanimously excited, citing lack of breadth in terms of models analyzed, and to much breadth (and lacking depth) in terms of the analysis performed.

---

### Decision · Program_Chairs · 2023-10-07

**Decision:**

Accept-Findings

**Comment:**

This paper presents a suite of probes for understanding when and where linguistic information (syntax, semantics, reasoning) is learnt in the embedded representations of MultiBERT models, including an information theoretic analysis supporting the specific formulation of the probes. The main concern shared by all reviewers is that the paper was often confusing and hard to follow. Some reviewers felt that treatment of highly related work was insufficient, some felt that the description of technical details was hard to follow, and some reviewers felt that the overall motivation/story of the paper was not clearly aligned with the provided results and analysis. If the paper is accepted, it should be substantially edited for clarity based on the feedback from the reviewers. After much clarification by the authors in the rebuttal, reviewers seemed to understand more, and found the results in the paper to be sound. While reviewers agreed that the topic was interesting and relevant, reviewers were not unanimously excited, citing lack of breadth in terms of models analyzed, and to much breadth (and lacking depth) in terms of the analysis performed.